# Encochleated Amphotericin B: Is the Oral Availability of Amphotericin B Finally Reached?

**DOI:** 10.3390/jof6020066

**Published:** 2020-05-18

**Authors:** Maria Aigner, Cornelia Lass-Flörl

**Affiliations:** Institute of Hygiene and Medical Microbiology, Medical University of Innsbruck, Schöpfstraße 41, A-6020 Innsbruck, Austria; cornelia.lass-floerl@i-med.ac.at

**Keywords:** amphotericin B, oral availability, antifungals, cochleates, MAT2203

## Abstract

As the oldest and for many decades the only available agent for the treatment of life-threatening invasive fungal diseases, amphotericin B (AmB) is known for its broad-spectrum fungicidal activity against a wide range of yeasts and molds. However, the main drawback of the present formulations remains its toxicity, the limited use to intravenous administration, and the higher costs associated with the better tolerated lipid formulations. The novel nanoparticle-based encochleated AmB (CAmB) formulation encapsulates, protects, and delivers its cargo molecule AmB in the interior of a calcium-phospholipid anhydrous crystal. Protecting AmB from harsh environmental conditions and gastrointestinal degradation, CAmB offers oral availability in conjunction with reduced toxicity. Matinas BioPharma, Bedminster, NJ is on the way to develop CAmB named MAT2203, currently undergoing Phase II clinical trials.

## 1. Introduction

Amphotericin B (AmB), a polyene antifungal agent, is known for its broad-spectrum fungicidal activity against a wide range of yeasts and molds including the protozoan parasite *Leishmania*. The physico-chemical properties of AmB (low solubility, tendency to self-aggregate in aqueous media, low permeability) lead to poor oral bioavailability [1]. Thus, amphotericin B deoxycholate (DAmB) and its lipid-based formulations (AmBisome^®^, Abelcet^®^, and Amphocil^®^) requires intravenous administration. The novel, nanoparticle-based encochleated formulation (encochleated amphotericin B, CAmB) prevents AmB from gastrointestinal degradation and thus enables oral availability [2]. Consisting of specifically arranged phospholipid bilayers and cations, the cochleate serves as drug delivery system that encapsulates, protects, and delivers hydrophobic substances such as AmB [3]. CAmB seeks to offer the benefit of reduced toxicities and low cost, further increasing accessibility and affordability especially in resource-limited settings. Collaborating with the National Institutes of Health/National Institute of Allergy and Infectious Disease (NIH/NIAID), Matinas BioPharma, Bedminster, NJ is on the way to develop CAmB named MAT2203 [4]. The U.S. Food and Drug Administration (FDA) granted MAT2203 as a Qualified Infectious Disease Product (QIDP) with Fast Track status with the potential for Orphan Drug Designation.

## 2. The Cochleate: Mode of Action

Cochleates are composed of a negatively charged lipid (i.e., phosphatidylserine) and a divalent cation (i.e., calcium) serving as binding agent to encapsulate hydrophobic, amphilic, negatively or positively charged moieties [5]. Through the interaction with the cations the phospholipid bilayers rearrange, form bilayer sheets, and result in cigar like spiral rolls with little or no internal aqueous phase [6]. Hydrophobic substances such as AmB, which aim to minimize their interaction with water, are incorporated in the multilayered lipid matrix at the interior of the calcium-phospholipid anhydrous crystal [7]. The external lipid layer offers protection for the carried “encochleated” drug from oxidation, environmental impacts, and enzymes and enables reception over the bloodstream [2]. Cochleates provide formulation stability for at least 3 months at 4 °C and for 7 days at 37 °C [8]. Although the precise mechanism is not fully understood yet, fluorescent-labelled investigations suggest that the cochleate edge, supported by calcium ions, fuses with the cell surface and releases its interior into the cytoplasm of the target cell [9,10]. Moreover, cochleates are engulfed by macrophages, which facilitates to reach the infected area. Once inside, as a response to the low calcium levels of the cytoplasm, the cochleate structure destabilizes, opens, and liberates the cargo molecule [5,11]. However, the quantity of cations in serum and mucosal secretions in vivo preserves the shape of the cochleate drug crystal [12]. Initial insights into altered pharmacokinetics and biodistribution of AmB after oral administration of CAmB are provided in a murine model of disseminated candidiasis [13]: Undetectable AmB plasma levels were found in 61% of the CAmB group (10 mg/kg orally) versus 44% in the DAmB (2 mg/kg intraperitoneally (ip) group. CAmB attained a defined MIC of 0.25 μg/mL in tissue by day 1–2; in contrast, DAmB with a delay after 3–5 days. While tissue levels of CAmB remained constant at 2–3× the MIC, DAmB levels accumulated up to 4–40× the MIC over time. Administration of doses up to 50 mg/kg/day CAmB orally for 14 days resulted in 100% survival of mice and normal, lesion-free histopathological findings in organs (kidneys, GI Tract, lungs, liver, and spleen) [5]. The NOAEL (no observable adverse effect level) was found to be at least 45 mg/kg/day and 90 mg/kg/day in dogs and rats for CAmB orally over a time period of 28 days, respectively [14].

## 3. In Vitro Data

In vitro, DAmB was shown to be highly hemolytic to human red blood cells at doses of 10 μg AmB/mL, while CAmB induced no hemoglobin release at concentrations up to 500 μg AmB/mL [8]. In contrast to DAmB, which demonstrated cell survival in only 36%, CAmB was non-toxic to mouse peritoneal macrophages at concentrations of 2.5 μg/mL [15]. Susceptibility data showed comparable minimal inhibitory concentrations (MICs) and minimal lethal concentrations (MLCs) of CAmB and DAmB against *Candida albicans* and a CAmB MIC of < 1 μg/mL against *Aspergillus fumigatus* [8,16]. Activity of CAmB against intracellular *Leishmania chagasi* amastigotes was equivalent to DAmB (ED_50_ of 0.017 μL/mL versus 0.021 μg/mL) in a macrophage infection model [15]. 

## 4. In Vivo Data

Various mouse models have provided data about the safety, efficacy, and tissue distribution of CAmb in comparison to other AmB formulations. An overview of obtained in vivo data is given in Table 1. In disseminated candidiasis, lowest doses of 0.5 mg/kg/day CAmB orally resulted in 100% survival equivalent to DAmB ip at a dosage of 2 mg/kg/day in mice [10]. In contrast, DAmB at 1 mg/kg/day ip showed a mortality of 30% by day 7. All administered doses of CAmB (0.5–5 mg/kg/day) led to a significant dose-dependent decrease of fungal tissue burden in kidneys and lungs. The highest efficacy was obtained at 2.5 mg/kg/day CAmB resulting in a 3.5 log reduction of CFUs (colony forming units) in the kidneys and complete clearance of the lungs comparable to DAmB at 2 mg/kg/day. In a murine model of cryptococcal meningoencephalitis using a strain of serotype A, *C. neoformans* CAmB (25 mg/kg/day) in combination with flucytosine (5FC) was equally effective as combination of DAmB (5 mg/kg/day) ip plus 5FC [17]. Superiority was shown to oral administration of fluconazole. Brain tissue levels of AmB and immunological profiles (cytokines as TNF-α, IL-1β and IL-6 and chemokines as CCL2 and CCL4) were comparable between both treatment regimens. The determination of fluorescent-labeled CAmB particles in brains of infected mice indicates adequate delivery of AmB via the encochleated oral formulation to the central nervous system. In contrast, fluorescence in uninfected mice brains was only marginal. Thus, infection appears to be a prerequisite for tissue penetration. However, the precise allocation of fluorescent-labelled signals to specific (immune) cells was not feasible. The efficacy of CAmB in comparison to DAmB (4 mg/kg/day) ip at different levels of immunosuppression was investigated in a murine model of invasive aspergillosis [16]. Mice were treated with cyclophosphamide at 150 mg/kg (low dose) and 200 mg/kg (high dose) with CAmB at 0 to 20 mg/kg/day and 0 to 40 mg/kg/day for 14 days, respectively. In the low dose group, CAmB at 10 mg/kg/day and 5 mg/kg/day led to a dose-dependent reduction in fungal tissue load (kidney, liver, and lungs) and mortality comparable to DAmB, respectively. Enhanced immunosuppression required 40 mg/kg/day and 20 mg/kg/day CAmB to gain survival of 90% and 70% versus 70% and 60% by days 4 and 14, respectively. While doses of ≥ 10 mg/kg/day reduced CFUs in tissues by a 2 to 3 log, doses ≥ 20 mg/kg/day almost eradicated fungal cells. In neutropenic mice CAmB at ≥ 5 mg/kg significantly increased survival and resulted in fungal clearance of all organs independently of the applied dosage [12]. CAmB reduced fungal load dose-dependently in kidneys and lungs but not in liver and spleen in died mice.

## 5. Phase I and II Trials

In healthy volunteers escalated doses of 200, 400, and 800 mg CAmB led to gastrointestinal adverse events (AE), most commonly nausea, in 6%, 38%, and 56%, respectively [18]. By the absence of any abnormal clinical laboratory findings, CAmB at a single dose was well tolerated especially at 200 and 400 mg. Efficacy, safety, tolerability, and pharmacokinetic data were obtained in an open-label, phase IIA study of oral CAmB in the treatment of chronic mucocutaneous candidiasis (CMC) in patients refractory or intolerant to standard non-intravenous therapy [19]. Four patients with STAT3 deficient Hyper IgE syndrome and CNC (*n* = 3) and chronic esophageal candidiasis (*n* = 1) were enrolled. CAmB led to clinical improvement of symptoms in 50%–85% at 400 to 800 mg/day. Adverse events grade 1 comprised nausea and dizziness. During a study period of up to one year, no signs of liver, kidney, or hematologic disorders were observed. Safety, tolerability, and efficacy of two dose regimens of MAT2203 (CAmB at 200 mg and 400 mg) compared with a single dose of fluconazole (150 mg) in the treatment of moderate to severe vulvovaginal candidiasis was evaluated in a proof-of concept phase II multicenter, randomized study [20]. The primary endpoint was to evaluate safety of MAT2203 at two dose regimens in comparison to fluconazole in 137 women. As secondary objective, clinical cure rate and mycological eradication was assessed in 79 women by day 12. By the absence of serious-, non-serious AE such as diarrhea, nausea, bacterial vaginosis, and urinary tract infections occurred in 22%, 27%, and 9% in patients receiving CAmB 200 mg, CAmB 400 mg and fluconazole, respectively. Clinical cure, mycological response and overall response (defined as clinical AND mycological response) was achieved in 52%, 36%, and 16% of patients at CAmB 200 mg; 55%, 32%, and 14% at CAmB 400 mg; and 75%, 84%, and 69% at fluconazole, respectively. Patients were recruited for the phase I/II “Encochleated Oral Amphotericin for Cryptococcal Meningitis Trial” (EnACT) in Uganda [21].

## 6. Novel Cochleate Structures

The non-proteolipidic non-lipidic Adjuvant Finlay Cochleate AFCo3 contains detoxified and purified MAMPs (microbe-associated molecular patterns), i.e., lipopolysaccharides from *Neisseria meningitidis* serogroup B, serving as vaccine adjuvants and immunopotentiator with the opportunity to add antibiotics/chemotherapeutic agents [22]. AmB antifungal and immunomodulatory activity was evaluated against *Sporothix schenkii* using AFCo3 as a vehicle [23]. MICs and MFCs (minimal fungicidal concentrations) were 0.25 μg/mL and 0.5 μg/mL for AFCo3-AmB and 1 μg/mL and 2 μg/mL for AmB, respectively. Compared to free AmB, AFCo3-AmB was less cytotoxic to peritoneal macrophages, demonstrated enhanced intracellular killing of phagocytized fungi and release of pro-inflammatory cytokines (IL-1ß, TNF-α and NO) and reduced hemolysis in murine erythrocytes. Besides normal blood chemistry findings, AFCo3-AmB was superior to AmB in reducing fungal burden in spleen and liver by day 7.

## 7. Discussion

Found in the 1950s, AmB was the first and for many decades the only available agent for the treatment of life-threatening invasive fungal diseases. Antifungal resistance is on the rise, ranging from azole resistance among *Candida* and *Aspergillus* species to echinocandin and multidrug resistance among especially *C. glabrata* species [24]. Although the longest time available, acquired resistance against AmB is still rare. However, its widespread use is hampered by its toxicity and costs associated with the better-tolerated lipid formulations. Due to the poor solubility in water and membrane permeability of AmB itself, detergents like the bile salt sodium deoxycholate need to be used for administration [11,25]. Efficacy, as well as partly toxicity, is mediated via the interaction of AmB with sterols of biomembranes of fungal and human cells (i.e., ergosterol and cholesterol) leading to increased cell permeability via pore-forming ion channels, cell leakage, rupture, and eventually cell death [11,26,27]. Immunomodulatory effects of AmB promote the release of pro-inflammatory mediators and contribute to toxic side effects such as acute infusion-related reactions (e.g., fever, chills, headache, nausea, vomiting, hypotension, arrhythmia) and nephrotoxicity [26,28,29]. Lipid-based formulations provide a better safety profile [25] but are still limited to parenteral administration. The packaging of AmB into cochleates yields the benefits of oral availability and potentially prevention of systemic exposure to the drug, thus reducing drug-related toxicity. The obtained in vitro and in vivo data appear to provide improved tolerability and safety profiles of CAmB compared to DAmB: (1) less cell-toxicity, which might be attributed to the lack of interaction between cells and free AmB through the encochleated formulation [8,15] and (2) less systemic toxicity due to the natural lipid composition of the outer layers of cochleates and stability of cochleates in serum and mucosal secretions [12], thus containing systemic interaction with the drug. As cochleates are engulfed by macrophages [30], which act at the site of infection, lower plasma levels to attain efficacious intracellular drug concentrations [12] are required. Moreover, the lack of drug accumulation in organs [13] may contribute to increased compatibility. In recent years, several nanotechnology-lipid-based drug delivery systems, including lipid conjugates, micelles, liposomes, ethosomes, and niosomes, nanoemulsions, SEDDSs (Self-Emulsifying Drug Delivery Systems), cubosomes, nanodisks, SLNs (Solid Lipid Nanoparticles and Nanostructured Lipid Carriers), and lipid–polymer hybrid nanoparticles have been investigated [11]. They all have in common to overcome the need for parenteral administration of AmB, to reduce its toxicity, but to preserve or even improve its efficacy. Moreover, several AmB formulations, comprising topical solutions for skin infections (AmB in 30% DMSO solution, 3% AmB cream (Anfoleish^®^), topical pulmonary (nebulized) and intrathecal AmB formulations, and AmB for injection are currently investigated at different stages of clinical trials [11,31]. To our knowledge encochleated AmB (CAmB/MAT2203) is the only available oral formulation, which has entered clinical development. Although efficacy and safety data gained from CAmB-animal models look promising, the phase II trial of CAmB (MAT2203) in women with VVC [20] remained far behind the expectations as both clinical and mycological responses were inferior to fluconazole. Higher dosages and longer duration of treatment might have resulted in better outcomes. As the potential of MAT2203 is seen in the treatment of invasive candidiasis and aspergillosis and for the prevention of invasive fungal disease in patients on immunosuppressive therapy [32], a comparative trial with fluconazole in mucosal infections is probably suboptimal. However, a planned open label phase II study to evaluate the safety and pharmacokinetics of oral encochleated Amphotericin B (CAmB/MAT2203) for antifungal prophylaxis in patients undergoing induction chemotherapy for acute myelogenous (AML) and lymphoblastic leukemia (ALL) has been withdrawn [33]. All in all, CAmB seems to offer an effective oral formulation associated with low toxicity. Further clinical trials are warranted to evaluate CAmB in humans and to compare its efficacy with other AmB formulations.

## Figures and Tables

**Table 1 jof-06-00066-t001:** Overview of in vivo data obtained from murine models and clinical trials.

Study	Treatment Regimen	Main Results				Reference
***Murine model of***						
• systemic candidiasis	CAmB vs. DAmB	• 100% survival at CAmB 0.5 mg/kg/day equivalent to DAmB 2 mg/kg/day.				[10]
		• dose-dependent reduction of CFU in kidney and lungs.				
• cryptococcal meningitis	CAmB + 5FC vs. DAmB + 5FC	• CAmB 25 mg/kg/day + 5FC equivalent to DAmB 5 mg/kg/day + 5FC.				[17]
		• Comparable brain tissue levels between CAmB and DAmB.				
• invasive aspergillosis	CAmB vs. DAmB	• Dose dependent reduction of CFU and mortality.				[16]
		• Survival rate of 70% at CAmB 20 and 40 mg/kg/day.				
*Clinical trials*						
• Phase IIA, CMC	CAmB	• Improvement of symptoms in 50%–85% at CAmB 400–800 mg.				[19]
		• AE: nausea, dizziness.				
		• No organ disorders.				
• Phase II, VVC	CAmB vs. fluconazole		CAmB 200 mg	CAmB 400 mg	Fluconazole 150 mg	[20]
		Clinical cure	52 %	55 %	75 %	
		Mycological response	36 %	32 %	84 %	
		Overall response	16 %	14 %	69 %	
		non-serious AE	22 %	27 %	9 %	

CAmB, encochleated amphotericin b (MAT2203); DAmB, amphotericin deoxycholate; CFU, colony forming unit; 5FC, flucytosine; CMC, chronic mucocutaneous candidiasis; VVC, vulvovaginal candidiasis; AE, adverse events.

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
