# Peer review of "Encochleated Amphotericin B: Is the Oral Availability of Amphotericin B Finally Reached?"

_jof, 2020, doi:10.3390/jof6020066_

Round 1
Reviewer 1 Report
Review Report – JOF – 800949
The present paper entitled “Encochleated amphotericin B: Finally reached oral availability of amphotericin B?” , a review manuscript, aimed to discuss the Cochleated Amphotericin B (CAmB) studies, and try to pave the way for the future use of such formulation against fungal infections.
The reviewing work was well conducted, although not too attractive for a reader. My feeling is that the authors, during the writing, missed to clearly stablish the goal and to connect the reviewer data in a fashion way to emphasizing the positive aspects of this new drug delivery system, CAmB. However, overall the manuscript sounds good and could be acceptable after the minor corrections suggested below:
1 – Please, revise the entire text for English mistakes, concerning the use of “commas”. The commas are missing in several sentences.
2 – The title has a grammar mistake. Maybe, it could be “Encochleated amphotericin B: Is the oral availability of amphotericin B finally reached?”.
3 – The sub reading 3. In vitro data, has to be In vitro data (in italic).
4 – Line 72. table 1, should be Table 1 (in capital letter).
5 – Line 104, CAmB is missing at the sentence. Should it not be “escalated doses of 200 mg, 400 mg and 800 mg of CAmB led….”
6 – The reference list has several important mistakes. Some titles are in capital letters, others not. Several Journal names are missing. Some journal’s titles aren’t in capital letter. A deep revision is mandatory.
Author Response
Response to Reviewer 1 Comments
Point 1: Please, revise the entire text for English mistakes, concerning the use of “commas”. The commas are missing in several sentences.
Response 1: commas added in line 12, 14, 26, 45, 50, 59, 72, 93, 94, 105, 106, 114, 120, 121, 147, 182, 183
Point 2: The title has a grammar mistake. Maybe, it could be “Encochleated amphotericin B: Is the oral availability of amphotericin B finally reached?”.
Response 2: The title has been revised as requested.
Point 3: The sub reading 3. In vitro data, has to be In vitro data (in italic).
Response 3: The sub-readings 3 and 4 have been changed into “in vitro”- and “in vivo”-data (Line 62 and 71)
Point 4: Line 72. table 1, should be Table 1 (in capital letter).
Response 4: revised as requested (Line 74)
Point 5: Line 104, CAmB is missing at the sentence. Should it not be “escalated doses of 200 mg, 400 mg and 800 mg of CAmB led….”
Response 5: revised as requested (Line 105)
Point 6: The reference list has several important mistakes. Some titles are in capital letters, others not. Several Journal names are missing. Some journal’s titles aren’t in capital letter. A deep revision is mandatory.
Response 6:
Reference 8: “candida albicans” revised into “Candida albicans” (capital letter and italic).
Reference 9: Authors and Journal name added.
Reference 12: Title revised into small letters.
Reference 13: Title revised into small letters.
Reference 14: Title revised into small letters.
Reference 15: “in vitro” revised into “in vitro” (italic), leishmania chagasi revised into “Leishmania chagasi” (capital letter and italic).
Reference 18: Title revised into small letters.
Reference 19: Journal name added.
Reference 20: Title revised into small letters.
Reference 21: Title revised into small letters.
Reference 23: “sporothrix schenkii” revised into “Sporothrix schenkii” (capital letter and italic). Journal name added.
Reference 24: Journal name added.
Reference 25: Journal name added.
Reference 27: Journal name added.
Reference 28: Journal name added.
Reference 29: Journal name added.
Reference 31: Title revised into small letters.
Reference 32: Title revised into small letters.
Reviewer 2 Report
The review article by Aigner et al summarised the literature reports and clinical studies regarding the encochleated amphotericin B (CAmB), aiming to provide some new understanding of the technology development. In general, the link to clinical information and translational progress would be informative and instructive for new researchers entering this field. However, the authors limited their scope to several formulations on the track of commercialisation, leaving other research studies without introduction and comparison. More research reports in the particle-based delivery of CAmB are suggested. Discussions of the pros and cons of current formulation under commercialisation with the newly dropped ones in literature are recommended. The review, in general, is in high quality and will attract the readers from its field, but thus recommended to be accepted after minor revision considering my above suggestions.
Author Response
Response to Reviewer 2 Comments
Reviewer 2: The review article by Aigner et al summarised the literature reports and clinical studies regarding the encochleated amphotericin B (CAmB), aiming to provide some new understanding of the technology development. In general, the link to clinical information and translational progress would be informative and instructive for new researchers entering this field. However, the authors limited their scope to several formulations on the track of commercialisation, leaving other research studies without introduction and comparison. More research reports in the particle-based delivery of CAmB are suggested. Discussions of the pros and cons of current formulation under commercialisation with the newly dropped ones in literature are recommended. The review, in general, is in high quality and will attract the readers from its field, but thus recommended to be accepted after minor revision considering my above suggestions.
Response: However, to our best knowledge, the precise mechanism of delivery of CAmB is not fully understood yet. We have precised the so far known modes of action (line 44 to 50).
We have added several nanotechnology-based drug delivery systems, which gained attention in recent years (line 165 to 171). Moreover, we have added several AmB formulations (topical, nebulized, intrathecal, for injection), which are currently investigated at different stages of clinical trials (line 171 to 175). To our knowledge, CAmB/MAT2203 is the only available oral formulation. Overall, they are too numerous to delve into deeply in this review, which aims to provide an overview of the encochleated formulation of AmB. However, we have added as a conclusion, that the efficacy of the encochleated formulation needs to be compared with other AmB formulations.